# Nutrient Requirements during Pregnancy and Lactation

**DOI:** 10.3390/nu13020692

**Published:** 2021-02-21

**Authors:** Marie Jouanne, Sarah Oddoux, Antoine Noël, Anne Sophie Voisin-Chiret

**Affiliations:** 1Université de Caen Normandie, UNICAEN, CERMN (Centre d’Etudes et de Recherche sur le Médicament de Normandie), F-14032 Caen, France; marie.jouanne@unicaen.fr; 2Laboratoire Dielen—Zone Produimer Port des Flamands, F-50110 Cherbourg-en-Cotentin, France; sarah-oddoux@dielen.fr (S.O.); antoine-noel@dielen.fr (A.N.)

**Keywords:** pregnancy, lactation, micronutrients, needs, supplements

## Abstract

A woman’s nutritional status during pregnancy and breastfeeding is not only critical for her health, but also for that of future generations. Nutritional requirements during pregnancy differ considerably from those of non-pregnant women. Thus, a personalized approach to nutritional advice is recommended. Currently, some countries recommend routine supplementation for all pregnant women, while others recommend supplements only when necessary. Maternal physiological adaptations, as well as nutritional requirements during pregnancy and lactation, will be reviewed in the literature examining the impacts of dietary changes. All of these data have been studied deeply to facilitate a discussion on dietary supplement use and the recommended doses of nutrients during pregnancy and lactation. The aim of this review is to evaluate the knowledge in the scientific literature on the current recommendations for the intake of the most common micronutrients and omega-3 fatty acids during pregnancy and lactation in the United States, Canada, and Europe. Taking into account these considerations, we examine minerals, vitamins, and omega-3 fatty acid requirements. Finally, we conclude by discussing the potential benefits of each form of supplementation.

## 1. Introduction

During pregnancy, women undergo a number of physiological changes [1,2] in order to achieve the normal development and health of the fetus. These changes also prepare the mother and baby for delivery.

The first change observed during pregnancy is weight gain. Following recommendations, for a woman with a normal weight (body mass index (BMI) between 19 and 24 kg/m^2^), gestational weight gain (GWG) should be between 11 and 16 kg. Physiological GWG is mainly due to fetus weight, the placenta, uterus, amniotic fluid, mammary gland, blood, and adipose tissue [3].

Moreover, hormonal changes are crucial throughout pregnancy [4]. On the one hand, there is an increase in the production of pre-existing hormones—mainly estrogens, progesterone, and prolactin—and the main producing tissues also change (the secretion becomes placental). On the other hand, specific hormones are synthesized by the placenta, like human chorionic gonadotropin (hCG). 

These hormones play a fundamental role in ensuring the proper course of pregnancy and their concentration evolves throughout pregnancy. For example, we can cite the role of progesterone in the thickening and the vascularization of the uterine lining in anticipation of the implantation of an embryo.

Other important changes are cardiac and hematological alterations [5]. Plasma volume increases gradually by more than 40% throughout a normal pregnancy. This expansion is greater than the increase in red blood cell mass—there is a decrease in hemoglobin concentration, hematocrit, and red blood cell count. The platelet count decreases at the end of pregnancy, although it usually remains within normal limits. To cope with this increase in volume, cardiovascular adaptation, with peripheral vasodilatation, a decrease in systemic vascular resistance, and an increase in cardiac output of about 40% is observed.

In a supine position, the pressure of the gravid uterus on the inferior vena cava causes a decrease in venous return to the right heart and hypotension. Venous pressure in the lower limbs increases for the same reason. This explains the frequency of edemas observed in the lower limbs.

Oxygen demand in pregnant women increases dramatically by 20 to 30%. Increased progesterone levels lead to an increased respiratory rate and increased ventilation. This hyperventilation is accompanied by some anatomic changes, such as diaphragmatic elevation supported by the gravid uterus or the extension of the lower ribs. This explains why many pregnant women feel short of breath.

Other changes include alterations of the gastrointestinal system with frequent nausea and vomiting during the first trimester, correlating with the hCG peak, gastroesophageal reflux, or constipation. The renal and urinary tracts are also affected by pregnancy [2].

A breastfeeding mother provides all the hydration and nutrients that a growing baby needs for the first 4–6 months of life. During pregnancy, the body prepares for lactation by stimulating the growth and development of branched lactiferous ducts and lactocyte-lined alveoli that secrete milk by creating colostrum. These functions are due to the actions of estrogen, growth hormone, cortisol, and prolactin. In addition, in response to progesterone, clusters of mammary alveoli bud from the ducts and dilate toward the chest wall. After childbirth, breastfeeding triggers the release of oxytocin, which stimulates myoepithelial cells to squeeze milk from the alveoli. Breast milk then flows to the pores of the nipple for consumption by the infant [6]. 

The nutritional needs of women increase during pregnancy and breastfeeding to support all of these changes, prepare the body for delivery and for breastfeeding, and to ensure the normal development of the fetus/baby.

Mainly provided by a balanced diet, micronutrients (i.e., vitamins and minerals) and omega-3 fatty acids are essential for many cellular and metabolic activities (cell differentiation, proliferation, hemoglobin production, transport oxygen, and mineralization, etc.). The current intake of vitamins, minerals, and omega-3 fatty acids from foods across Europe limit the risk of severe deficiencies [7]. However, some deficiencies have been highlighted as particularly affecting pregnant women, such as vitamin D or iron deficiencies [8,9,10]. This is why pregnant women should be vigilant about their food intake so that they maintain adequate levels of micronutrients. Vitamins, minerals, and omega-3 fatty acids play an important role during pregnancy: ensuring the appropriate progress of a normal pregnancy in order to support the mother through the common discomforts of pregnancy or to prevent pregnancy complications. In addition, even though lactation is considered successful when the breastfed baby gains an appropriate weight, it is recommended that women continue to take a prenatal vitamin daily while breastfeeding.

Faced with changes related to pregnancy, the needs of the fetus, and the preparation of the body for breastfeeding, the nutritional needs of pregnant women can be adapted either by the implementation of an adapted and balanced diet or by supplementation. Specific nutritional intakes can also correct some common clinical signs of pregnancy.

In this review, we provide an overview of the current recommendations for the intake of the most common micronutrients and omega-3 fatty acids during pregnancy and lactation in the United States, Canada, and Europe. Therefore, we look at the requirements for minerals, vitamins, and omega-3 fatty acids. We conclude with a discussion about the potential benefits of each form of supplementation.

## 2. Methods 

In the first step, in order to identify relevant studies, a comprehensive search process was conducted on the following scientific electronic databases: PubMed, Google Scholar, SciFinder, and Web of Science, without language restrictions, to identify eligible studies published between 1981 and 2020. In parallel, a Cochrane Database search was undertaken using the following search words: ‘‘minerals’’ or “vitamins” or “omega-3 fatty acids” associated with “pregnancy” or “lactation” or “supplementation/supplements” or “nutritional needs”. Additional records were also identified, allowing us to specify the data that concerned a personalized approach to nutritional advice.

In the second step of the screening process, we followed the PRISMA (preferred reporting items for systematic reviews and meta-analyses) principles in developing the review and, after identification of records through database searches, duplicates were removed. 

In the third step, the exclusion criteria were manuscripts written in a language other than English or French; webinars, blogs or podcasts; and articles involving topics that were clearly irrelevant. Years considered and publication status were used as criteria for eligibility. Inclusion criteria were: (i) publication in a peer-reviewed journal; (ii) human studies; (iii) studies examining the impacts of nutrients on pregnancy and lactation; and (iv) studies examining the impacts of changes in dietary habits. 

In the fourth step, the selected articles were read in full and additional articles identified from their references were also reviewed. A total of 85 manuscripts were selected and included in the present review.

The characteristics of the included original studies are shown in Figure 1.

To establish a scientific judgment, it is necessary to set reference values [12]:Recommended dietary allowance (RDA): the average daily dietary intake level that is sufficient to meet the nutrient requirement of nearly all healthy individuals in a group.Adequate intake (AI): a value based on observed or experimentally determined approximations of nutrient intake by a group (or groups) of healthy people—used when an RDA cannot be determined.Tolerable upper intake level (UL): the highest level of daily nutrient intake that is likely to pose no risk of adverse health effects to almost all individuals in the general population. As intake increases above the UL, the risk of adverse effects increases.Estimated average requirement (EAR): a nutrient intake value that is estimated to meet the requirement of half the healthy individuals in a group.

## 3. Results and Discussion

### 3.1. Role of Minerals, Vitamins, and Omega-3 Fatty Acids in Pregnancy—Needs and Benefits/Risks of Supplementation

#### 3.1.1. Iron and Vitamin B9

Iron plays an important role in the production of hemoglobin and for the transport of oxygen; therefore, in the face of increased blood mass, fetal growth, and the development of appendages, including the placenta, the iron requirements of pregnant women are markedly increased (22–27 mg/day) (see Table 1). In response to these increased needs, during pregnancy, the intestinal absorption capacities of iron are also increased (globally) from 10 to 40% at the end of pregnancy [13,14,15,16,17]. It should be noted that vitamin C can help with the intestinal absorption of iron, but that tea and coffee can decrease it (due to the presence of polyphenols). Some studies advise consuming tea between meals instead of during a meal. A one-hour time interval between a meal containing iron and the consumption of tea attenuates the inhibitory effects on iron absorption [18,19].

Moreover, anemia due to iron deficiency is common in pregnant women and 2 to 5% of women are diagnosed during the first trimester of pregnancy, while 10 to 20% of women are diagnosed in the third trimester (in industrialized countries). Anemia is considered moderate and severe when hemoglobin (Hb) levels are between 7 and 9 g/dL and less than 7 g/dL, respectively.

A diagnosis of iron deficiency anemia can be made when the threshold values are as follows:Hb level <11 g/dL in the first and third trimesters;Hb level <10.5 g/dL in the second trimester;ferritin level <30 μg/L: insufficient iron reserve.

Otherwise, a postpartum hemoglobin level less than 9 g/dL is a factor associated with a higher risk of post-traumatic stress disorder [20]. Overall, maternal iron intakes decrease the risk of having a low birth weight or a premature baby and increase the average birth weight of infants [13].

Vitamin B9 (natural food folate), also known as folic acid (the synthetic form), is metabolically inactive. Enzymatic reductions enable the conversion of folic acid to dihydrofolate (DHF), and then tetrahydrofolate (THF). THF is then reduced to obtain the biologically active L-methylfolate. L-methylfolate is a methyl donor necessary for DNA replication and RNA synthesis, DNA methylation, and to regulate homocysteine metabolism [21].

Folic acid is chemically more stable than folate and has better bioavailability. Data on the bioavailability of food folate vary, but it is estimated that food contributes 50% of all bioavailable folic acid. Considering this bioavailability difference, the concept of dietary folate equivalents (DFEs) is used: 1 μg of DFE is equivalent to 1 µg of food folate, 0.6 μg of folic acid in fortified foods, or 0.5 µg in supplements [22,23,24].

Pregnancy is a common cause of folate deficiency—especially in multiple pregnancies or when pregnancy is complicated by vomiting. Folate deficiency can be responsible for some pregnancy complications, such as primarily neural tube defects (NTD), including spina bifida and anencephaly [22]. It is also important to emphasize that a 5 mg daily dose is recommended for women with pregestational diabetes [25]. 

Therefore, vitamin B9 requirements for pregnant women are 400 μg/day [22] (see Table 1) and it is recommended that women take folic acid supplementation for at least 3 months before pregnancy. In actual fact, it would be better if each woman regularly consumed folic acid supplements throughout her fertile life.

However, the intake of folate can be higher than the recommended dietary allowance and, although folate is believed to be non-toxic, the potential adverse effects of an excessive intake of folic acid must be mentioned. Studies suggest that high folic acid intake may, under certain conditions, promote cancer, interact with medications, and impair fetal development. Studies in mice have shown more sinister impacts, suggesting that high levels of folic acid have serious damaging consequences by causing epilepsy and liver damage [26].

Iron and vitamin B9 supplementation is recommended by the WHO in women only when anemia is proven by a complete blood count (CBC) with measurement of the hemoglobin (Hb) level. The determination of serum ferritin concentration can also be performed to assess iron stores and diagnose an iron deficiency.

Thus, in the case of severe to moderate anemia, the WHO recommendation is a daily oral iron supplementation with 30 mg to 60 mg of elemental iron in order to prevent maternal anemia, puerperal sepsis, low birth weight, and preterm birth [13]. Although the requirement is most important in the last trimester, it is important to build up iron stores early and avoid high doses later, so that the higher intake recommendation is distributed throughout the pregnancy [27].

To reduce the risk of neural tube defects, WHO recommendations indicate that all women in the periconceptional period (eight weeks before and eight weeks after conception) should take a folic acid supplement (400 μg folic acid daily) [13]. Current evidence suggests that folic acid supplementation in the periconceptional period, either alone or in combination with other vitamins and minerals, can prevent neural tube defects [28]. Therefore, supplementation is recommended in the periconceptional period (3 months before gestation).

Daily supplementation

Evidence of the effects of daily iron and/or folic acid intake was obtained from a Cochrane review of 61 trials in low-, middle-, and high-income countries [29]. Daily oral iron and folic acid supplementation with 30 to 60 mg of elemental iron (60 mg of elemental iron equivalent to 300 mg of ferrous sulfate heptahydrate, 180 mg of ferrous fumarate, or 500 mg of ferrous gluconate) and 400 μg of folic acid is recommended for pregnant women to prevent maternal anemia, puerperal sepsis, low birth weight, and premature deliveries. 

Intermittent supplementation

Evidence of the effects of intermittent iron and folic acid supplementation was obtained from a Cochrane review, which included 27 trials from 15 countries; however, only data from 21 trials (involving 5490 women) were used for the review’s meta-analyses [29]. Intermittent oral iron and folic acid supplementation with 120 mg of elemental iron (120 mg of elemental iron equivalent to 600 mg of ferrous sulfate heptahydrate, 360 mg of ferrous fumarate, or 1000 mg of ferrous gluconate) and 2800 μg of folic acid once a week is recommended for pregnant women to improve the outcomes of pregnancy for both mothers and newborns if daily iron intake is not acceptable due to side effects and in populations where the prevalence of anemia in pregnant women is less than 20%.

#### 3.1.2. Calcium

Calcium participates in the mineralization of the fetal skeleton, especially during the third trimester. The skeleton of a full-term baby contains approximately 30 g of calcium, and three-quarters of this mineral content is deposited during the last trimester of pregnancy. As a result, maternal calcium needs to increase, especially from the third trimester (the need for calcium varies from 1000 to 1200 mg/day) (see Table 1) [13,16].

To meet these increased needs, the intestinal absorption of calcium increases very early in pregnancy. In addition, the vitamin D supplementation recommended in the seventh month of pregnancy in some cases (see below) promotes this intestinal absorption of calcium. Low calcium intake can worsen the severity of last-trimester bone loss and the risk of developing pre-eclampsia.

Calcium supplementation is only recommended by the WHO for low calcium intake populations to reduce the risk of pre-eclampsia.

The WHO indicates that calcium can be used for the relief of pregnant women’s leg cramps. The WHO Guideline Development Group (GDG) agreed that calcium and magnesium are unlikely to be harmful in the dose schedules evaluated in the studies they reviewed (i.e., the studies by Zhou et al. [30], Hammar et al. [31,32], and Sohrabvand et al. [33]). However, the GDG specified that further research into the etiology and prevalence of leg cramps in pregnancy, and the role (if any) of magnesium and calcium in symptom relief, is needed [13].

In addition to the WHO recommendations, two studies have published the results of calcium supplementation during pregnancy:O’Brien et al. [34] highlighted that intakes of dietary calcium <1000 mg/day, particularly for pregnancies in winter (where vitamin D stocks are low due to low sunshine) were associated with increased bone resorption. They concluded that calcium supplementation improves bone resorption at the end of pregnancy, particularly for winter pregnancies.A recent Cochrane review [35] on the effects of calcium supplementation during pregnancy in preventing hypertensive disorders and related problems has identified 24 studies that evaluated the effects of high-dose supplementation (≥ 1000 mg/day versus placebo) or low-dose supplementation (<1000 mg/day versus placebo). The authors conclude that high-dose calcium supplementation may reduce the risk of pre-eclampsia and preterm births, especially for populations with low-calcium diets. It also appears that supplementation with low doses would reduce the risk of pre-eclampsia and hypertension. They moderate these conclusions by specifying that additional research with larger and better-quality clinical trials is necessary not only to confirm these effects, but also to verify that this supplementation does not have any adverse effects, particularly for the fetus.

In populations with low calcium intake, daily supplementation (1.5–2.0 g of elemental calcium) is recommended for pregnant women to reduce the risk of pre-eclampsia [34]. Evidence of the effects of calcium supplementation on outcomes other than hypertension/pre-eclampsia was obtained from a systematic Cochrane review [36].

In addition, the WHO recommends using calcium to relieve cramps in the lower extremities during pregnancy (oral calcium administration versus no treatment) [34]. Administration of calcium twice daily for two weeks has been compared to no treatment in a small study. A low level of evidence suggests that women receiving calcium treatment are less likely to have leg cramps after treatment. This is why women should be encouraged to consume dairy products even if they do not have an insufficient calcium intake. Mineral waters rich in calcium (> 150 mg/L) can also be offered if dairy products cannot be consumed (e.g., digestive disorders).

#### 3.1.3. Magnesium

During pregnancy, serum magnesium levels gradually decrease, reaching low values during the last trimester and increasing after childbirth (variations compared to physiological hemodilution) [37,38]. The concentration of magnesium in the blood of the umbilical cord is higher than maternal magnesium levels, which means that active transport occurs through the placenta, where 50% of the average amount of dietary magnesium is absorbed. In addition, to meet the needs of the fetus, magnesium requirements increase during pregnancy because renal magnesium excretion is enhanced by about 25% [39]. The magnesium requirements of pregnant women are difficult to define (around 350 mg/day) because normo-magnesemia does not rule out magnesium deficiency: additional tests have revealed that the lower limit of the reference interval of magnesemia is not optimal. Indeed, if the basis of an optimal serum magnesium concentration over 0.80 mmol/L is used, the majority of pregnant women suffer from a magnesium deficiency when only the serum concentration is taken into account [40]. 

Magnesium deficiency has been implicated in the occurrence of hypertensive disorders, gestational diabetes mellitus, preterm labor, and intrauterine growth restriction [41].

The WHO recommends the use of magnesium to relieve cramps in the lower extremities during pregnancy (study with oral magnesium administration versus placebo) [34]. In three studies, women in the intervention group received 300 to 360 mg of magnesium daily, divided into two or three doses. Studies have compared the persistence or occurrence of leg cramps in different ways, so the results could not be aggregated. The small amount of available evidence suggests that, on the one hand, women receiving magnesium are more likely to see a 50% decrease in the number of leg cramps, and on the other hand, oral magnesium has little to no effect on the occurrence of potential side effects, including nausea, diarrhea, flatulence, and bloating.

Although it has been shown that European people consume lower amounts of magnesium than is recommended [42] and that, during pregnancy, magnesium requirements increase, the effect of supplementation during pregnancy remains inconsistent, so there is no justification for systematic maternal magnesium supplementation. A Cochrane review concluded that there is not enough high-quality evidence to show that dietary magnesium supplementation during pregnancy is beneficial [38] and that careful orientation of food choices should be sufficient to ensure adequate intakes. Therefore, the WHO does not recommend magnesium supplementation.

However, if women’s needs are around 350 mg/day during pregnancy, 80% of pregnant women have an intake of less than 300 mg/day (intake also reduced due to the magnesemia that occurs during pregnancy). As a result, neuromuscular consequences resembling cramps appear. Supplementation (as soon as possible) with 200 mg/day is then effective [43,44,45].

#### 3.1.4. Iodine

Thyroid homeostasis, especially in pregnant women and fetuses, is essential for the development of brain tissue, the acquisition of intelligence, and learning. The main sources of iodine in the diet come from foods that contain it (e.g., fish, seafood, and dairy products) and certain additives that are fortified or rich in iodine (e.g., cooking salt). However, health experts recommend that pregnant women avoid certain types of fish and seafood during pregnancy because they have a high risk of contamination with parasites, germs, or toxins.

Moreover, during pregnancy, iodine requirements are increased by approximately 50% due to maternal thyroid stimulation (by hCG), an increase in renal iodine clearance and iodine transfer to the fetus for the synthesis of fetal thyroid hormones from the second trimester. 

The WHO recommendation for iodine intake during pregnancy is 220–250 µg/day [16,46]. 

Particular situations expose pregnant women to a high risk of deficiency: living in a particularly deficient area; smoking; having pregnancies that are too close together; consuming a specific diet (e.g., veganism); andsuffering nausea and vomiting, thereby reducing food intake [16].

#### 3.1.5. Zinc

Zinc is essential for many biological processes including, for example, cell division, protein synthesis and growth, and nucleic acid metabolism. During pregnancy, zinc deficiencies may lead to congenital malformations, low birth weight, intrauterine growth retardation, and preterm delivery [47]. The zinc requirements of pregnant women are slightly increased (11 mg/day) (see Table 1); however, zinc is mainly present in meat, fish, and seafood. As such, food intake alone may, therefore, be insufficient during pregnancy. Zinc deficiency is common worldwide, particularly in developing countries. In European countries, there is no severe deficiency [48]; however, pregnant women should monitor their dietary intake because plasma zinc concentration is an important determinant of pregnancy outcomes [49].

Zinc supplementation in pregnant women is only recommended as part of rigorous research and with context-specific recommendations.

Evidence was obtained from a Cochrane review, which included 21 trials involving more than 17,000 women [50]. The Cochrane database reported a 14% relative reduction in preterm birth with zinc supplementation compared to a placebo group. However, the authors noted that these results were obtained primarily in studies of women from low-income households and this has some relevance in areas of high perinatal mortality. In addition, they noted that there was not enough evidence to show that zinc supplementation in women results in other clinically relevant outcomes [50].

Moreover, pregnant women taking iron supplementation should be even more vigilant because it has been shown that iron decreases zinc absorption. However, this effect is only observed with high concentrations of iron and when zinc and iron are given in solution. Thus, it is advised that iron supplements should be taken between meals [51]. In addition, zinc increases the absorption of dietary folates and thus contributes to the prevention of folate deficiencies. 

#### 3.1.6. Vitamin D

Vitamin D is available in two forms: D2 and D3. Vitamin D2 (ergocalciferol) is the form found in plant-based sources. Vitamin D3 (cholecalciferol) is the form found in dietary animal sources and is also the form produced by human skin (vitamin D3 is synthesized, starting from 7-dehydrocholesterol, by the epidermis upon exposure to ultraviolet B radiation)

Vitamin D is converted into 25-hydroxyvitamin D (25-OH-D) by the liver and then into the active metabolite 1,25-dihydroxyvitamin D (1,25(OH)_2_D) by the kidney, among other organs. This active form ensures the mineralization of mineralized tissues (bones, cartilage, and teeth) during and after growth in order to contribute, alongside parathyroid hormone, to maintaining calcium homeostasis.

Vitamin D, particularly 25-OH-D, is the subject of an active transplacental transfer (correlation between the plasma concentration of 25-OH-D in the mother and that in the umbilical cord) [52]. On the other hand, there is no correlation between the maternal rates of 1,25-(OH)_2_-D in maternal and umbilical cord blood: 1,25-(OH)_2_-D crosses the placental barrier poorly. The regulation of its synthesis is, therefore, specific to the fetoplacental unit.

In adults, circulating levels of 25-OH-D are particularly low during the autumn and winter with low sun exposure. Pregnant women have a vitamin D deficiency at the end of pregnancy, especially during winter or early spring. There is a relationship between this poor vitamin status and the frequency of early and late neonatal hypocalcemia accidents, as well as early deficient rickets (partly linked to the initial poor vitamin status). The most deficient pregnant women can develop symptomatic osteomalacia during pregnancy.

Therefore, it appears essential to ensure that pregnant women have the correct vitamin status: the recommended intakes have been set at 10–15 µg/day (400–600 international units (IU)/day) (see Table 1) [53]. However, this intake is often insufficient, particularly, during the third trimester and during the months of low sunshine. When it is only administered in the third trimester, 1000 IU/day are then necessary to obtain concentrations of 25-OH-D within normal limits in the mother and in the cord blood. The same results can be obtained with a single dose of 200,000 IU administered at the start of the seventh month (no higher loading dose should be used due to potential toxicity). This intake has reduced the frequency of neonatal hypocalcemia [54]. Information on the most effective and safe dosage, the optimal dosing regimen (daily, intermittent, or single doses), the timing of initiation of vitamin D supplementation, and the effect of vitamin D when combined with other vitamins and minerals is also needed in order to inform recommendations [55].

Vitamin D supplementation is not recommended by the WHO for pregnant women to improve the maternal and perinatal outcomes of pregnancy. Nevertheless, new trials demonstrate the effects of vitamin D supplementation alone or with other nutrients during pregnancy on maternal and newborn health. Evidence was obtained from a systematic Cochrane review, which included 30 trials involving 7033 women [55]. According to the WHO, for pregnant women with a suspected vitamin D deficiency, vitamin D supplements may be given at the current recommended nutrient intake of 200 IU (5 μg) per day. This may include women in populations where sun exposure is limited [56]. 

However, it should be noted that supplementation with vitamin D compared to no intervention or a placebo:Probably reduces the risk of pre-eclampsia, the risk of gestational diabetes, and the risk of having a baby with low birthweight (less than 2500 g).May make little or no difference to the risk of preterm birth.In terms of maternal adverse events, vitamin D supplementation may reduce the risk of severe postpartum hemorrhage, although it should be noted that this result is based on findings from a single trial and was an unexpected finding that has not been documented before by any other study.

In addition, a recent study [57] shows that vitamin D supplementation should be a mandatory basic care recommendation for all women, especially women of childbearing age and pregnant women. Moreover, pregnant women have a vitamin D deficiency at the end of pregnancy (third trimester), especially when this occurs in winter or early spring. Consequently, it appears essential to ensure that pregnant women have the correct vitamin status. Even if the IU number and the dosing schedule (daily or once) are not determined, it should be recommended to achieve at least a serum 25-OH-D level of 20 ng/mL.

#### 3.1.7. Vitamin A

The term vitamin A includes the free and esterified retinols present in food, their metabolites produced in the body, which are responsible for their biological activity (retinol and retinoic acids), as well as the pro-vitamin carotenoids (β-carotene, α-carotene, and β-cryptoxanthin). In order to take into account the incomplete conversion of pro-vitamin carotenoids to retinol, the vitamin activity of these compounds is expressed as the retinol equivalent (RE) according to the following formulas:1 μg RE = 1 μg retinol = 6 μg β-carot = 12 μg other provitamin A carotenoids

(Vitamin A requirement can be met with any mixture of preformed vitamin A and provitamin A carotenoids that provides an amount of vitamin A equivalent to the reference value in terms of μg RE/day [58]).

The essential nature of vitamin A is due to the role of retinoic acid in the regulation of gene expression and in cell differentiation. Retinol is also essential for vision and contributes to the functionality of the immune system, the health of the mucous epithelia (especially of the eye), and growth.

In France, for example, the average consumption of RE by pregnant women is around 1000 µg/day, which is close to the recommendations (see Table 1). Fetal plasma and, in particular, hepatic concentrations remain fairly constant, regardless of the vitamin status of the mother: insufficient intakes do not, therefore, result in a particular fetal or neonatal consequence. Thus, the risk of deficiency is low.

Caution should be exercised with vitamin supplements as there is a risk of hypervitaminosis A being teratogenic: an upper safe limit of 3000 µg/day has been set by the Scientific Committee on Food (SCF), confirmed by the European Food Safety Authority (EFSA) [59]. Vitamin A supplementation is only recommended for pregnant women in areas where vitamin A deficiency is a serious public health problem, with the aim of preventing night blindness.

Evidence was obtained from a systematic Cochrane review of 19 vitamin A supplementation trials (with or without other supplements) versus no vitamin A supplementation (or placebo/other supplements) involving more than 310,000 women [60].

#### 3.1.8. Other B-Group Vitamins 

The recommended intakes of most B-group vitamins are increased above non-pregnant values as shown in Table 1. The increases are based on evidence of the higher maternal requirements, as well as fetal and placental deposition of vitamins [27].

B-group vitamin deficiency alone is rare; it mainly occurs in association with the deficiencies of several B-group vitamins.

However, the WHO recommends taking vitamin B6 (pyridoxine) to relieve nausea in early pregnancy. An average level of evidence obtained from two trials (in one, 25 mg of vitamin B6 was administered orally every eight hours for three days and in the other, 10 mg of vitamin B6 was administered by orally every eight hours for five days) shows that vitamin B6 likely reduces nausea symptoms, but a low level of evidence suggests that there is little to no effect on vomiting [61].

In general, vitamin B6 supplementation is not recommended in pregnant women for the purpose of improving the maternal and perinatal outcomes of pregnancy.

Evidence was obtained from a Cochrane review that included four trials involving approximately 1646 pregnant women: a low level of evidence suggests that oral pyridoxine supplementation may have little or no effect on preeclampsia; a moderate level of evidence shows that vitamin B6 probably provides some relief from nausea during pregnancy [61].

#### 3.1.9. Vitamins E and C Supplementation 

Vitamin E and C supplementation is not recommended by the WHO for pregnant women in order to improve the maternal and perinatal outcomes of pregnancy.

Evidence was obtained from two systematic Cochrane reviews that included 17 trials from low-, middle-, and high-income countries [61,62].

It should be noted that a high level of evidence shows that vitamin E and C supplementation is associated with an increased risk of abdominal pain during pregnancy.

#### 3.1.10. Omega-3 Fatty Acids 

Polyunsaturated fatty acids (PUFAs) are necessary for the optimal functioning of the brain. A food deficiency that modifies the membrane composition, particularly that of omega-3, is a source of dysfunction at the metabolic, physiological, and behavioral level. Clinical studies have also established that low consumption of omega-3 or low plasma levels of docosahexaenoic acid (DHA) are associated with cognitive and behavioral disorders during development. The accumulation and incorporation of DHA in the brain takes place mainly during the perinatal period when neural networks are established. While the cellular and molecular mechanisms are not yet well understood, the growing data explain multiple actions.

As fundamental constituents of membranes, PUFAs can play on their physicochemical properties and, therefore, on the proteins contained therein. As precursors of lipid mediators, they are involved in many regulatory processes, particularly inflammation. They are also ligands for nuclear receptors and, thus, participate in the regulation of genes involved, especially in lipid and carbohydrate metabolism.

Therefore, the dietary intake of PUFAs during pregnancy is of obvious importance for brain development. Indeed, the spectacular accumulation of arachidonic acid (AA) and DHA in the brain from the sixth month of pregnancy allows us to consider that these are essential elements for the maturation of the brain, the period during which the development of neuronal extensions takes place—the establishment and stabilization of synapses and myelination.

However, the brain’s desire for omega-3, especially during neurological development [64], justifies paying attention to the nutritional intakes of pregnant women of omega-3, more particularly DHA. The incorporation of DHA in the brain has been evaluated at 3 mg/day during the last trimester of pregnancy.

In the Elfe study, the food intake of pregnant women was assessed using a self-administered food frequency questionnaire in 14,257 women during their stays in hospital maternity units. This questionnaire focused on consumption during the last three months of pregnancy. This questionnaire makes it possible to assess the contributions observed. Under-consumption situations affect between 50% and 75% of women for linoleic acid and DHA, and more than 75% of women for alpha linolenic acid (ALA) and eicosapentaenoic acid (EPA) [65].

Seafood rich in omega-3 and data from observational studies suggest a possible link between eating seafood during pregnancy and the reduced risk of developing certain health problems, such as pre-eclampsia, premature delivery, and low birth weight.

Current evidence suggests that omega-3 supplementation is associated with a reduced risk of preterm delivery and a modest increase in birth weight.

It is sometimes impossible to know if the mother-to-be’s diet is balanced or whether it will be difficult to change her eating habits. Two 170-gram servings of low-mercury fish and seafood are recommended per week for pregnant women. Consuming more may pose a risk of mercury toxicity, although the absolute risk is low. Alternatively, an adequate supply of omega-3 fatty acids can be derived from supplements such as fish oil and certain prenatal vitamins. Fish oil capsules are almost devoid of mercury and other harmful compounds like polychlorinated biphenyls and can be used to increase omega-3 fatty acids in the diet. 

Prospective studies in pregnant women who consumed the recommended fish intake, or received fish oil supplements, generally demonstrate a beneficial effect on the neurodevelopmental outcomes of their offspring [66]. At present, there is insufficient data to recommend supplementation with omega-3 fatty acids for the sole purpose of reducing the risk of preterm delivery or preventing perinatal depression. Nevertheless, the option of omega-3 supplementation may be an interesting prospect to ensure the growth of the fetus, which needs to receive omega-3 via the placenta. Supplementation may also be considered in nursing mothers. As such, it is important to follow the nutritional recommendations between pregnancies, in order to replenish DHA for the next baby. 

### 3.2. Micronutrients and Omega-3 Fatty Acid Requirements for Breastfeeding Women 

Scientific evidence on which micronutrient intake recommendations for breastfeeding are based is sparse compared to that published in relation to pregnancy. Lactation is considered successful when the breastfed baby gains an appropriate amount of weight [17].

Weight loss during breastfeeding generally does not affect the quantity or quality of breast milk, but maternal deficiencies of magnesium, vitamin B6, folate, calcium, and zinc have been reported during breastfeeding [67,68]. The fat-soluble vitamins A, D, and K and the water-soluble vitamins C, B1, B6, B12, and folates are secreted in breast milk and their content is reduced in breast milk in the event of vitamin deficiency in the mother [69,70,71]. Fortunately, these vitamin deficiencies in breast milk respond to maternal supplementation. In addition, the levels of calcium, phosphorus, and magnesium in breast milk are independent of the serum levels and the mother’s diet [72]. Maternal factors such as stress, anxiety, and smoking can reduce milk production, but nutritional and caloric values of breast milk are not impacted.

A recent study was interested in comparing the effectiveness of maternal vitamin D3 supplementation with 6400 IU/day alone and both maternal and infant supplementation with 400 IU/day [73]. The results show that maternal vitamin D supplementation at 6400 IU/day safely provides breast milk with enough vitamin D to meet the needs of infants and offers an alternative strategy to guide the supplementation of infants.

As we have seen previously, omega-3 intake while breastfeeding is also important for brain development in the first 2 years of life.

The incorporation of DHA in the brain has been evaluated at 5 mg/day during the lactation period. Thus, it is estimated that, during the first 6 months, the brain, which represents 8% of the body weight at this age, accumulates 905 mg of DHA since birth in comparison to 977 mg for the rest of the body during the same period.

The value of DHA intake during gestation (interventional study with a 400 mg/day intake) in women from the 16th week of gestation until birth is studied by a visual acuity test adapted to infants [74]. Several clinical studies in premature or full-term infants have led to questions about the neurosensory consequences of using a milk substitute too low in ALA and completely free of DHA. These children were shown to have a delay in the development of visual functions associated with a low blood level of DHA compared to that of breastfed children of the same age [74,75]. 

The insufficient conversion rate of PUFAs in newborns compared to their cerebral needs led to the proposal of recommendations for a DHA intake of 100 mg/day during the first year of life [76]. 

More generally, there are few articles identifying the micronutrient needs of breastfeeding women and the table below summarizes the data associated with these articles (Table 2) [77,78,79].

## 4. Limitations

Pregnancy requires a healthy diet that includes an adequate supply of energy, protein, vitamins, and minerals to meet the increased needs of the mother and the fetus. However, for many pregnant women, dietary intake of fruits, vegetables, meat, and dairy products is often insufficient to meet these needs and can lead to micronutrient deficiencies. In the resource-poor countries of sub-Saharan Africa, south central Asia, and southeast Asia, maternal undernutrition is widespread and is recognized as a key determinant of poor perinatal outcomes [80]. However, an objective understanding of the personal needs and contributions of all essential vitamins and minerals for optimizing maternal and fetal health during the prenatal period is limited [81].

Additionally, most studies and reviews of the literature dealing with maternal nutrition and birth outcomes have addressed the problem by examining specific nutrients in isolation. At some level, this is necessary for a thorough study of the complex issues involved. However, nutrient deficiencies are generally found in populations of low socioeconomic status, where they are more likely to result in multiple rather than single deficiencies [82], while studies that address and aggregate the larger picture of multiple nutrient intakes or deficiencies are rare [83].

Therefore, the results of multiple micronutrient supplementation need to be taken into account. However, interventional studies on pregnant women are limited to very specific issues. Therefore, ideally, observational studies on large and homogenous cohorts would be required to assess the value of supplementation. Nutritional status before conception and during pregnancy and infancy appears to influence the risk of disease in adulthood. Our objective was to review the current knowledge on the role of micronutrients in early nutrition programs and their implications during pregnancy and lactation for women and children.

In 2020, a recommendation updated and replaced the WHO recommendation in the WHO antenatal care guidelines published in 2016. The recommendation was derived from trials using multiple micronutrient supplements containing 13–15 micronutrients (including iron and folic acid) and the widely available United Nations International Multiple Prenatal Micronutrient Preparation (UNIMMAP), which contains 15 micronutrients, including 30 mg iron and 0.4 mg folic acid. Many multiple micronutrient supplements contain 30 mg or less of elemental iron, and the WHO recommends prenatal iron and folic acid supplements containing 60 mg of elemental iron in populations where anemia is a serious public health problem (prevalence of 40% or more) [84]. 

Therefore, countries should take into account their population size and the distribution of anemia, its nutritional determinants (i.e., iron deficiency), as well as the complex magnitude and distribution of low birthweight and its component parts (i.e., preterm, small for gestational age, or a combination of these) [85] when performing any research according to this recommendation.

## 5. Conclusions

Nutrition counselling is essential for all women during pregnancy. A woman’s nutritional status affects her health, but also the outcome of pregnancy and the health of the newborn baby. Nutritional requirements during pregnancy differ considerably from those of non-pregnant populations. 

The literature is rich on this subject and our conclusions were based on a sufficient level of evidence. We can therefore consider two situational categories: systematic additions or special situations that may justify supplementation or curative or preventive treatment of maternal or fetal pathologies.

The current recommended daily intakes seem very far from physiological realities and, consequently, overestimate the real needs of pregnant women. Today, all of the evidence indicates that coping mechanisms allow well-nourished, healthy women, who have a varied diet available to them, to carry a normal pregnancy to term without any other resources than those provided by their food consumption and the spontaneous increases that occur during pregnancy.

Nevertheless, it is recommended that a personalized approach to nutritional counselling is adopted, which takes into account women’s access to food, socio-economic status, race, ethnicity, and cultural food choices, as well as BMI. In addition, many of the recommendations relate to uncomplicated pregnancies, so adjustments should be made when complications, such as gestational diabetes, arise.

## Figures and Tables

**Figure 1 nutrients-13-00692-f001:**
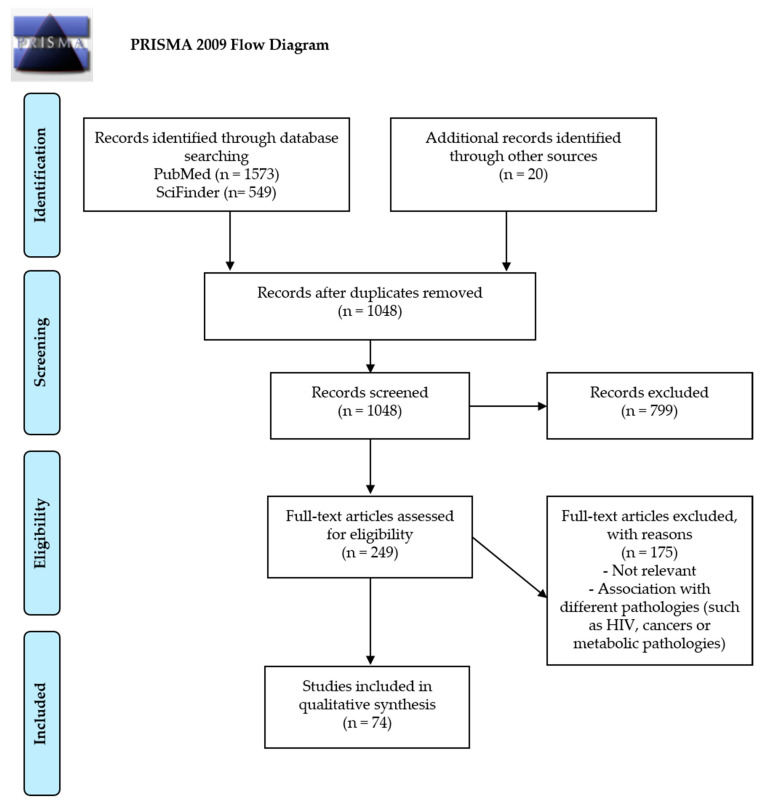
PRISMA (preferred reporting items for systematic reviews and meta-analyses) 2009 flow diagram from Moher, D. et al. [11].

**Table 1 nutrients-13-00692-t001:** Recommended dietary allowances (RDA) for non-pregnant women and estimated average requirements (EAR), RDA, and tolerable upper intake level (UL) for pregnant women. Adapted from Medeiros et al. [63], Kominiarek et al. [17], and Allen [27].

Nutrient	RDA, Adult Non-Pregnant Women	EAR, Pregnancy	RDA, Pregnancy	UL, Pregnancy	Justifications
Vitamin A (μg/day)	700	550	770	3000	Regulation of genome expression and in cell differentiation
Vitamin D (μg/day)	15	10	15	100	Mineralization of the fetal skeleton and decreased risk of hypocalcemia accidents and symptomatic osteomalacia
Vitamin E (mg/day)	15	12	15	1000	–
Vitamin K (μg/day)	90	–	90	none	–
Vitamin B1 (mg/day)	1.1	1.2	1.4	none	–
Vitamin B2 (mg/day)	1.1	1.2	1.4	none	–
Vitamin B3 (mg/day)	14	14	18	35	–
Vitamin B6 (mg/day)	1.3	1.6	1.9	100	Relieve nausea in early pregnancy
Vitamin B9 (μg/day)	400	520	600	1000	Decreases the risk of spina bifida and other neural tube defects
Vitamin B12 (mg/day)	2.4	2.2	2.6	none	–
Vitamin C (mg/day)	75	70	85	2000	–
Calcium (mg/day)	1000	800	1000	2500	Mineralization of the fetal skeleton Prevents pre-eclampsia
Iodine (μg/day)	150	160	220	1100	Maintenance of thyroid homeostasis
Iron (mg/day)	18	22	27	45	Decreases the risk of having a low birth weight or a premature baby
Magnesium (mg/day)	320	290	350	350	Involvement in the occurrence of hypertensive disorders, gestational diabetes mellitus, preterm labor, or intrauterine growth retardation
Phosphorus (mg/day)	700	580	700	3500	–
Selenium (μg/day)	55	49	60	400	–
Zinc (mg/day)	8	9.5	11	40	Involvement in cell division, protein synthesis and growth, nucleic acid metabolism

**Table 2 nutrients-13-00692-t002:** Micronutrient requirements in lactating women.

	Recommendations	Justifications for Breastfeeding
Calcium	1000 mg/day	Maintenance and production of breast milk
Magnesium	390 mg/day	Muscle relaxant Prevention of constipation
Zinc	19 mg/day	Participation in postpartum healing
Vitamin C	130 mg/day	Stimulation of immune functions
Vitamin D	10 µg/day = 400 IU ^a^/day	Important contribution to obtain good quality milk
Vitamin A	10,000 IU/day or max 25,000 IU/week or unique intake 200,000 IU	Only in deficient populations, as soon as possible after childbirth, but not more than 8 weeks afterwards
Iron	60 mg/day	Prevention of maternal anemia For 3 months after postpartum
Vitamin B9	400 µg/day	
Omega–3	100 mg/day of DHA ^b^ during the 1st year of the newborn’s life	Newborn brain development

^a^ IU: international units; ^b^ DHA: docosahexaenoic acid.

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
