# Peer review of "Nutrient Requirements during Pregnancy and Lactation"

_nutrients, 2021, doi:10.3390/nu13020692_

Round 1

Reviewer 1 Report

General comments

This review manuscript collects the results of a search for recommended amounts of micronutrients, vitamins, and omega-3 fatty acids during pregnancy/lactation, and information on the significance of supplements. Although the abstract and conclusion discuss the importance of an individualized approach to nutritional advice, the main text does not specifically address an individualized approach.

This review lacks a proper procedure for the systematic review. It is unclear what is the low level of evidence and what is not.

If the authors are preparing this review as a systematic review, they should first revise their methods. Why not include Cochrane Library in the search from the beginning?

It is unclear how many records were found against the search terms for all databases at the beginning, how the studies were selected, and what is the method for evaluating the evidence level. In the case of the low level of evidence or the high level of evidence, the corresponding citation should be clearly stated.

Table 1 in this manuscript states that it follows the reference Nutrition Recommendations in Pregnancy and Lactation, but some values have been changed. The authors should clearly state the rationale for these changes in the main text.

It is also necessary to explain what is the definition of the required amount, the recommended amount, the upper and the lower intake limits of each nutrient.

Specific comments

Lines 37-39

During pregnancy, estrogen, progesterone, and prolactin are produced by the placenta. It should be stated that the major producing tissues change.

Lines 173-174

I understand that one literature recommends 750 and the other 1200. The previous sentence explains that the demand increases in late pregnancy. Why is there a recommended dose of 1000 in non-pregnancy, but the recommendation drops to 750 in pregnancy?

Lines 238-239

Assembly is an inappropriate term. Please change to “DNA replication and RNA synthesis”

The harmful effects of excessive intake of folic acid supplements should also be mentioned.

Lines 267-273

Write a quantitative value for the correlation between Vitamin D levels in maternal blood and cord blood. Please cite the references.

Lines 285-293

Please cite the references.

Line 309

The term "genome expression" is inappropriate.

Please change it to Gene expression.

Lines 492-494

“In populations with low calcium intake, daily supplementation (1.5-2.0 g of elemental calcium) is recommended for pregnant women to reduce the risk of pre-eclampsia. “

Is this text taken from reference 52? If so, please include it in the paragraph above and summarize it. If there are different references, please list them.

Lines 580-581

Please provide references of average level of evidence (two trials) and low level of evidence.

Reviewer 2 Report

Thank you for sharing with us this comprehensive review on the intake of micronutrients during pregnancy and lactation.

I recommend to combine in section 3 (of the discussion) the role and needs with the benefits of the micronutrients, so you don't duplicate the discussion on each micronutrient.

It is also very important to cite the recommendations of the professional medical international societies in different disciplines: OBG, Endocrinology, Nutrition.

Please find some remarks and corrections:

Line 2- It is better to change in the title the phrase "Nutrients contribution".

Introduction:

Line 28-I would use another word instead of "must", preferably "recommended".

Line 34-the hormone is HCG and not HCG beta.

Lines 81-82: The sentence is not clear enough. What  do you mean?

Discussion:

Lines 131-133: It is better to use the term trimesters.

Line 188- How should pregnant women monitor their zinc intake?? It is not clear.

Line 206: For vitamin B9-folic acid, the OBG literature usually recommends a 400 mg daily intake. Please use a reference for this recommendation. You mention it later in the manuscript, so the final message is confusing.

It is also important to emphasize that in women with pregestational diabetes, a 5 mg daily dose is recommended. 

In addition, add the recommendation to take folic acid supplementation at least 3 months before pregnancy. Actually. it is better for each woman to take it regularly during her fertile life.

Lines 225-6: 25-OH-D may be insufficient, not always. No correlation is known to milk or milk products. If you find a reference for this statement, please add it.

For the vitamin D paragraph, it is important to discuss whether to take a daily dose or once-weekly/once-monthly dose, as recommended in the past in non-pregnant subjects.

Line 248: The formula is not clear enough.

Line 342: Change the title of the table to Table 2 and not 1. Iron is mentioned twice in two different rows. Put a legend to the table, and explain what is the abbreviation DHA.

Line 343: Maybe the title is a little bit misleading as you show later that not always the vitamins have benefits during pregnancy.

Line 345: Mention that you refer to iron supplementation.

Line 353: It is important to mention that some recommend folic acid supplementation already 3 months before gestation.

Lines 364-366: This concept was already mentioned in a previous paragraph.

Line 360: Ref. 45-Check whether it is the right reference; In the Reference list this is probably ref. 46 and not 45.

Line 370: Check that ref. 46 is the correct one.

Lines 447-449: Does this sentence relate to the previous one? It is not clear.

Conclusions:

Lines 579-583: Probably this conclusion is true for women in the Western world. You should elaborate on it more before the "Conclusions". It is better to compare in detail the nutrition in different parts of the world.

References:

References 2 and 6 are the same, and also 15 and 35.

Reference 36 include actually two different references??? Check the same for ref. 37-includes 3 references?? The same for Ref. 64. 

Round 2

Reviewer 1 Report

I think that the methods section has been revised, and references have been added and duplicates removed, but it seems that there have been no essential improvements.

 This manuscript attempts to describe the physiological changes that occur during pregnancy and lactation period compared to pre-pregnancy, and to review on the required intakes of vitamin, mineral, and omega-3 fatty acid. As the authors acknowledge, nutritional needs are not the same around the world, so it would be helpful to the reader if the authors first stated in the abstract which region in the world is the target of this manuscript.  As I previously pointed out, the term “individualized approach” should not be used because this review focuses on the difference between pregnancy/ lactation and pre-pregnancy and does not say anything about how nutrition recommendations should be personalized.

Particularly, section 3 is redundant and difficult to read, and should be reorganized.

As suggested by Reviewer 2, it would be better to summarize the role of nutrients, intake, and supplements for each nutrient in one sequence to avoid redundancy and make it easier to understand. For example, in terms of iron, the demand for iron increases from the early to late stages of pregnancy. Then how should iron supplements be taken for this?  Is it better to increase the intake of iron during pregnancy? In the current manuscript, iron and folic acid supplements are described in a mixed manner, which leaves these questions unanswered.

Also, for each nutrient, please describe whether it is possible to establish an estimated average requirement (EAR) and recommended dietary allowance (RDA), or whether it is only a reference amount. For nutrients for which it is possible to establish an EAR and RDA, it is recommended that the EAR and RDA for the same age group of non-pregnant women be noted, and then the amount added during pregnancy be summarized. Also, please specify if there is a difference between the 1st, 2nd, and 3rd trimester. Please describe all the values of recommended amounts, reference amounts, and upper limits when possible in the table. It is helpful for reader to obtain the summary of the text from the table at a glance.

The current version is not suitable for publication. More significant improvement is needed.

Reviewer 2 Report

Dear colleagues,

Thank you for all the efforts.

However, very extensive editing of the English language is still needed.

I recommend to send it again for editing the language and the style.

Just for some examples in the beginning of the manuscript:

Line 23 it should be written-"a potential benefit of each supplementation"

Line 28- "During pregnancy, women present...."

Lines 40-41-the addition "and major producing...." should be rephrased.

Lines 41-42: "On the other hand, specific hormones are ....."

etc. etc.
